

# Evidence of preliminary prognosis of appearance of catastrophic earthquake and strong tsunami in the region of Tarapacá, Chile

**Mazova Raissa.Kh.[1,2], Van Den Bosch Jorge F.[3], Baranova Natalia.A.[4], Oses Gustavo A.[5]**

*[1]Nizhny Novgorod State Technical University named after R.E. Alekseeva, Nizhny Novgorod, Russia e-mail: raissamazova@yandex.ru*
*[2] Moscow Institute of Physics and Technology (MIPT) 141701 Moscow, Russia e-mail: raissamazova@yandex.ru*
*[3]Engineering Center Mitigation Natural Catastrophes Faculty of Engineering. 1240000 Antofagasta, Chile. e-mail address: Jorge.VanDenBosch@uantof.cl*
*[4]Auriga, 6/6 Nartova st., office 829, 603104 Nizhny Novgorod, Russia.*
*e-mail address: Natalia.baranova@inbox.ru*
*[5]Engineering Center Mitigation Natural Catastrophes Faculty of Engineering. 1240000 Antofagasta, Chile. e-mail address: Gustavo.oses@uantof.cl*

**Abstract.** Are analyzed the catastrophic seismic events near Chilean coast and generated by them tsunami 1 April 2014 to north of Iquique with magnitude 8.2. It is noted that event occurred 1 April 2014 was in fact predicted in work (Mazova and Ramirez 1999), in which there were analyzed all strongest Chilean tsunamigenic earthquakes with sources near the Chilean coast. Analysis of catastrophic earthquakes and tsunamis in given region, localization of source of historical earthquakes and character of generated by them tsunami waves permitted authors in that time to make a conclusion about possibility of repeated 20 catastrophic earthquake and tsunami in near 10-20 years. The events near Iquique and Arica city in April 2014 are in this time period. Thus, the evidences, presented in this work, support preliminary prognosis made by authors in 1999.

**1. General characteristics of seismic activity off the coast of Chile.** As was previously predicted by a number of authors see, e.g., (Mazova and Ramirez 1999), (Mazova and Soloviev1994) seismic activity around the perimeter of the Pacific Ocean 25 will increase substantially by the end of the 20th and the beginning of the 21st centuries. Indeed, there was a series of catastrophic earthquakes accompanied by a tsunami: in the Indian Ocean near Sumatra on December 26, 2004 with a magnitude of 9.1, earthquake and tsunami on November 15, 2006 in the Kuril-Kamchatka region, a catastrophic earthquake with a magnitude of 8.8 that occurred in the middle part of Chile on February 27, 2010, a tsunamigenic catastrophic earthquake around Japan on March 11, 2011 with a magnitude of 9.2 and the last event - a mega-earthquake and tsunami in northern Chile 30 on April 1, 2014 with a magnitude 8.2. The nature of the generated tsunami waves, for each of these events, their distribution and behavior in the coastal zone, have been analyzed in sufficient detail in the literature see, e.g., (Pararas-Carayannis 2010), (Ramirez et al. 1997). The need to consider the earthquake source of a more complex form, adequate to the implementation of aftershocks during an earthquake, was shown in our previous works. (R.K Mazova et al. 2014), (Lobkovsky et al. 2006).

It is well known that in subduction zones the strongest earthquakes with M ≥7 usually generate tsunamis. An analysis of 35 catastrophic earthquakes shows that the destruction of infrastructure and population loss in coastal areas are mainly not due to



movements of the earth's surface during the earthquake itself, but because of the tsunami that followed it, e.g., (Murty 1977), (Mazova et al. 1983). Tsunami is a danger not only for the nearby coast, but also for the coasts located at a considerable distance from the area of their generation. Most of the sources of tsunamigenic earthquakes in the Pacific seismic belt are located on the terraces of the continental slope of the deep-water trench (Lobkovsky L and Lobkovsky B 1984), (Lobkovsky L 1988) and their properties are essentially determined by the type of subduction zone. According to modern concepts of geotectonics (Lobkovsky L1988), (Lobkovsky L et al. 2004 ) , there are two types of subduction zones: the Chilean type and the Mariana type. The Chilean type is characterized by a deep-sea trench and a strong coupling between the continental and oceanic lithospheric plates during the subduction process, and seismic activity in the Chilean-type subduction zones leads to strong tsunamis. Examples of Chilean-type subduction zones are the areas of the Kuril-Kamchatka and Japanese deep-sea trenches. Seismic activity in the Chilean subduction zone leads to the emergence of strong tsunamis that have repeatedly attacked the coast of Chile and caused great damage and human casualties, e.g. (Mazova and Ramirez 1999), (Murty 1977), (Mazova et al. 1983). Thus, historical records indicate that a series of catastrophic tsunamigenic earthquakes (Mazova and, Ramirez 1999), (Mazova et al. 1983) occurred off the coast of Chile. Chile is located in one of the most seismically active areas of the world with 15-30 weak shocks daily in northern Chile. This high level of seismic activity is due to the geological structure of northern Chile, where a deep-water trench (up to 3000 m deep) is located near the coast. However, there is also a very smooth continental slope that extends 150 km from the coast to the trench, thus forming the terrace.

It is well known that the tsunamigenic earthquakes that occur near the northern coast of Chile pose a danger to local shores and shores thousands of kilometers beyond this coast. The tsunami that occurred during the Chilean earthquake in 1868 reached the shores of New Zealand. Similarly, the tsunami caused by the Chilean earthquake in 1960 reached the shores of the Kamchatka Peninsula, the Kuril Islands and Japan. Historical records indicate that in the XIX century a number of catastrophic tsunamigenic earthquakes occurred on the northern coast of Chile (Pararas-Carayannis 2010), (Emily et al. 2014). One of them happened on August 13, 1868 with a maximum magnitude of M = 8.5 in the southern Peru region, with an earthquake source extended to the northern coast of Chile, near the city of Arica, which caused a large tsunami, later called as Arica tsunami. The epicenter of the earthquake was less than 100 km from the coast, on the terrace of the deep-water trench and in almost all coastal points where the tsunami was recorded, it began with the withdrawn of water from the coast, followed by a wave train in which the second wave was the most destructive (Pelinovsky & Mazova 1992). Presumably, the movements in the seismic source were directed downwards and were accompanied by horizontal movements of the keyboard blocks. Downward movement in the area of the seismic source was directed to the coast of Chile. The tsunami waves generated by this earthquake reached the maximum height of the runup on the shore up to 21 m. Eyewitnesses, survivors, described it as a tsunami: "From the ocean with a thundering noise, a huge wall of phosphorescent and foaming water washed over" or "The Ocean has washed over the shore in the form of a terrible wave ... carrying a trial on its crest." Also, with the source off the northern coast of Chile, on May 10, 1877, a destructive earthquake with a magnitude of M = 8.8 occurred, which was accompanied by a catastrophic tsunami. In the literature, this tsunami is referred to as the Iquique tsunami. In the city of Iquique, the heights of the waves on the coast reached 4.8 m, and at various points on the coast of South America, tsunami waves attack the shore



with a height of 24 m. Entire blocks of cities were washed away and destroyed. In Iquique alone, 30 people died. In the 20th
      century, three catastrophic earthquakes accompanied by a tsunami also occurred on the coast of Chile, of which two were
      located in central and northern Chile. Thus, on August 16, 1906, an earthquake with a magnitude of 7.8 occurred in the central
      northern part of the Chilean coast. The greatest damage was in the city of Valparaiso. The maximum heights of tsunami waves
      along the coast reached 3.5 m. Runup process ashore proceeded calmly, in the form of coastal flooding. On May 22, 1960, a

destructive mega-earthquake with a source in the southern part of middle Chile with a magnitude of M = 9.5 occurred. The
      maximum rise of water on the coast of Chile reached 25 m. In July 1995, in the north of Chile, near the city of Antofagasta, a
      destructive tsunamigenic earthquake with a magnitude of M = 8.1 occurred, the sea reached 130cm above the corresponding
      level for that moment ( Ramirez et al. 1997) while  in Caleta Blanco, a place located 60 km south of the city of Antofagasta
      the  Nazca plate, had the highest penetration 2 m, (Fig.1) (Mendoza 1997) and also in this place the highest intensity is recorded,

the mountain  throw rocks of great size and weight, happily that area was unpopulated, the sea registered the highest runup of
      245 cm measured from the average level of the sea. Despite the modest size, this tsunami was reported to Pacific-wide, and
      reached 300 cm amplitude on the coast of Hiva Oa, Las Marquesas Islands, which sank two small boat and inundated 40,000
      km$^2$ of land (Ramirez et al.1997).

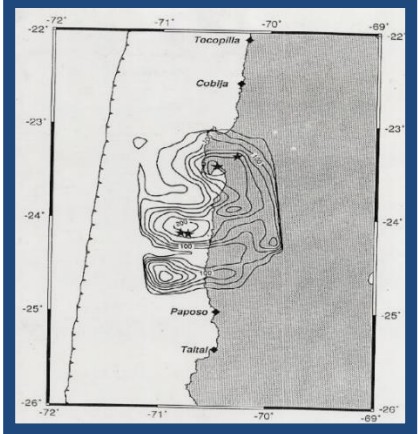

**Fig.1. Greater displacements of the Nazca plate during the 1995 earthquake in Antofagasta  (Mendoza 1997)**
      In the XXI century, the strongest earthquakes occurred in the middle and northern part of the coast of Chile at the beginning
      of the century (Table 1). Thus, on February 27, 2010, a catastrophic earthquake with a magnitude of 8.8 occurred near the
      coast of Chile, generating a powerful tsunami. ( Mazova and Ramirez 1999), (Pararas-Carayannis 2010), (Hamlington et al.
      2011), (Paula et al.2010),  The earthquake source (35.909° S, 72.733° W) was located in the sea at a depth of 35 km under the

earth's crust 17 km of the coastal settlements of Curanipe and Cobquecura, 90 km of the capital Bio-Bio Concepción, 150 km
      northwest of Concepcion and 63 km southwest of Cauquenes. The Chilean earthquake caused a powerful tsunami — twenty
      minutes after an earthquake, a sea wave two meters in height hits the coast of Chile. April 1, 2014 in northern Chile, near the
      Chilean coast, an earthquake of 8.2 magnitude occurred.



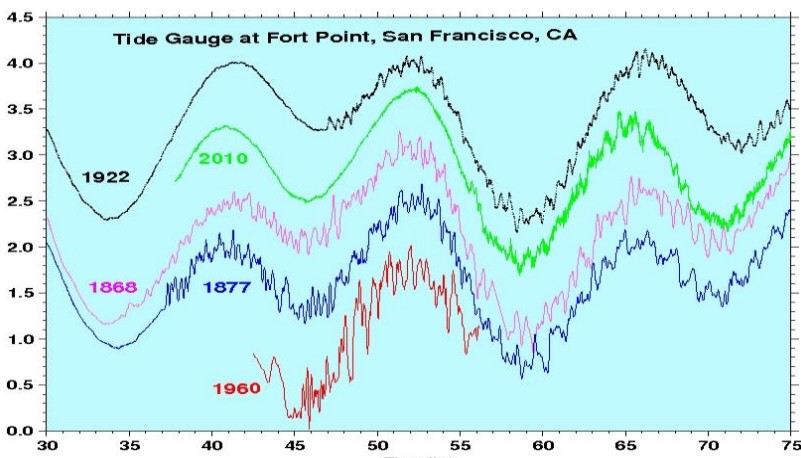

**Fig.2. Barrientos S. Wards S. 2008. (Private communication)**

This earthquake occurred in an area of historical seismic gap, called the northern Chilean or Iquique seismic gap, there in the last 574 years there have been 5 major earthquakes on average every 114 years, all with records of destructive tsunamis for the cities of northern Chile and southern Peru, as well as some destructive results across the Pacific, example of these tsunamis

originated in the north of Chile are 1868 and 1877 of which there are tide gauge records of the time in the station Fort Point California USA, (Fig.2), in addition you can see other tsunamis generated in the north central and south of Chile, the size of the wavelength varies from the highest rate corresponding to the 1960 earthquake to the lowest corresponding to the 1922 earthquake.

**Table 1. Earthquakes and tsunamis on the coast of Chile for 2010-2014.**

| Date and time | Geographic coordinates | Magnitudes | Region, contributed magnitudes and comments | Nearby cities |
|---|---|---|---|---|
| 2014-04-03 02:43:14 UTC | 20.518°S 70.498°W | 7,7 | Northern Chile. This earthquake is an aftershock of the M 8.2 subduction zone earthquake that occurred April 1, 2014. The M 8.2 event triggered a tsunami with measured heights near 2 meters along the northern Chile and southern Peru coasts. Since the M8.2 event, 47 aftershocks ranging from M 4.2 to this M 7.8 event have occurred, including a M 6.4 on April 2. | 1. 49km (30mi) SW of Iquique, Chile<br>2. 177km (110mi) N of Tocopilla, Chile<br>3. 227km (141mi) S of Arica, Chile<br>4. 269km (167mi) NW of Calama, Chile<br>5. 509km (316mi) SSW of La Paz, Bolivia |





| Date | Coordinates | Mag | Description | Distances |
|------|-------------|-----|-------------|-----------|
| 2014-04-01 23:57:57 UTC | 19.898°S 70.924°W | 6,9 | Northern Chile. | 1.  89km (55mi) WNW of Iquique, Chile<br>2.  170km (106mi) SSW of Arica, Chile<br>3.  221km (137mi) SSW of Tacna, Peru<br>4.  253km (157mi) S of Ilo, Peru<br>5.  476km (296mi) SW of La Paz, Bolivia |
| 2014-04-01 23:46:46 UTC | 19.642°S 70.817°W | 8,2 | Northern Chile.<br>The April 1, 2014 M8.2 earthquake in northern Chile occurred as the result of thrust faulting at shallow depths near the Chilean coast.<br>The April 1 earthquake occurred in a region of historic seismic quiescence – termed the northern Chile or Iquique seismic gap. Historical records indicate a M 8.8 earthquake occurred within the Iquique gap in 1877, which was preceded immediately to the north by an M 8.8 earthquake in 1868. | 1.  95km (59mi) NW of Iquique, Chile<br>2.  139km (86mi) SSW of Arica, Chile<br>3.  190km (118mi) SSW of Tacna, Peru<br>4.  228km (142mi) SSE of Ilo, Peru<br>5.  447km (278mi) SW of La Paz, Bolivia |
| 2014-03-16 21:16:30 UTC | 19.925°S 70.628°W | 6,7 | Northern Chile. This earthquake is considered the last precursor of the 8.2º major earthquake on April 1, 2014 in this area [19] | 1.  60km (37mi) WNW of Iquique, Chile<br>2.  164km (102mi) SSW of Arica, Chile<br>3.  216km (134mi) S of Tacna, Peru<br>4.  244km (152mi) N of Tocopilla, Chile<br>5.  460km (286mi) SW of La Paz, Bolivia |
| 2010 March 16 02:21:58 UTC | 36.124°S, 73.147°W | 6,7 | Central Chile (offshore). | 70 km (45 miles) NNW of Concepcion, Chile<br>110 km (70 miles) WNW of Chillan, Chile<br>160 km (100 miles) NNW of Los Angeles, Chile<br>390 km (240 miles) SW of SANTIAGO, Chile |
| 2010 March 11 14:39:44 UTC | 34.259°S, 71.929°W | 6,9 | Central Chile. Damage reported at Rancagua. A small tsunami was recorded with wave heights (peak-to-peak) of 16 cm at Valparaiso and 29 cm at San Antonio. | 105 km (65 miles) W of Rancagua, Chile<br>130 km (80 miles) N of Talca, Chile<br>140 km (85 miles) S of Valparaiso, Chile<br>145 km (90 miles) SW of SANTIAGO, Chile. |


| 2010 February 27 06:34:14 UTC | 35.909°S, 72.733°W | 8,8 | Central Chile (offshore). At least 523 people killed, 24 missing, about 12,000 injured, 800,000 displaced and at least 370,000 houses, 4,013 schools, 79 hospitals and 4,200 boats damaged or destroyed by the earthquake and tsunami in the Valparaiso-Concepcion-Temuco area. A Pacific-wide tsunami was generated. Tsunami wave heights in centimeters (above sea level) were recorded at more than 60 tidal stations in the Pacific Ocean. | 95 km (60 miles) NW of Chillan, Chile 105 km (65 miles) NNE of Concepcion, Chile 115 km (70 miles) WSW of Talca, Chile 335 km (210 miles) SW of SANTIAGO, Chile |

Seismic activity in the Chilean subduction zone leads to the emergence of strong tsunamis that have repeatedly attack the coast of Chile and led to great destruction and human casualties.

**2. Earthquake and tsunami 01.04.2014 near the northwestern part of the Chilean coast**

It is important to note that the earthquake was preceded by an increasing seismicity and followed a South-North direction pattern (Emily et al.2014); in this regard, during the month of March 2014, shortly before the earthquake occurred on April 1, 14 precursor earthquakes occur in the area of rupture; 3 earthquakes of magnitude 4.5º to 4.9º, 7 earthquakes of magnitude 5.1º to 5.8º and finally very close to April 1 there are 4 earthquakes of magnitude 6.2º to 6.7º. This is probably the historical model of precursor earthquakes that precede a great earthquake in this area. Something similar happened in this same area before the great earthquake of August 13, 1868, the story indicates the following: "August began with frequent seismic movements that kept the neighbors in distress and caused the amusement of the sailors, anchored to a mile of location, when they saw them launch into the streets with screams and terror", the author of this story, refers to the observations of the sailors anchored off the port of Arica some two weeks before the great earthquake of 1868 on the north coast of Chile .(Urzúa 1969).

It is analyzed catastrophic seismic events off the northern part of the Chilean coast on April 1, 2014. Around the city of Iquique, the earthquake had a magnitude of 8.2, the epicenter was at the point of 20.518° S and 70.498° W in the area of the seismic gap, called the northern Chilean or Iquique seismic gap. Tsunami waves were recorded on the coast of Chile in the cities:

Pisagua - 1.7 m; Yunin - 1.7 m; Iquique - 1.6 m; Punta Negra-1.6 m; Alto Hospicio - 1.6 m; Chucumata - 1.3 m; Arica-0.7 m.

These data, as well as data on aftershocks that were freely available, were the basis of the present study. The calculation was carried out in the Pacific Ocean in a square of $60^0$-$90^0$ W and $10^0$-$60^0$ S (grid size 1798x2995), bathymetry with a resolution of 1". Numerical simulation was performed using the finite-difference scheme. (Sielecki and Wurtele 1970).

To implement the task, two scenarios of earthquake source formation and subsequent tsunami source generation were considered. Data on the formation of a seismic source were taken from Internet resource ECDM_20140402_ Chile_Earthquake 2014.



**Table 2. Data on aftershocks for the first day of the earthquake on April 1, 2014.**

| Time | Magnitude | Depth | Epicenter |
|---|---|---|---|
| 1 April 23:56:47 | 5.7 | 10.1 km. | 92 km. NW from Iquique |
| 23:58:00 | 6.2 | 18.1 km. | 88 km. NW from Iquique |
| 2 April 00:03:12 | 5.7 | 10.1 km. | 92 km. NW from Iquique |
| 00:06:44 | 5.5 | 10.0 km. | 88 km. NW from Iquique |
| 00:24:45 | 5.2 | 10.6 km. | 66 km.WNW from Iquique |
| 00:33:45 | 5.4 | 5 km | 52 km. W from Iquique |
| 00:37:49 | 5.4 | 20.1 km. | 51 km. WNW from Iquique |
| 01:20:58 | 5.3 | 10.0 km. | 111 km. NW from Iquique |
| 01:29:41 | 5.2 | 10.0 km. | 75 km. NW from Iquique |
| 02:52:25 | 5.0 | 10.0 km. | 108 km. SW from Arica |
| 03:40:16 | 5.7 | 10.1 km. | 92 km. NW from Iquique |
| 04:19:48 | 5.1 | 10.0 km. | 93 km. WNW from Iquique |
| 04:46:18 | 5.8 | 10.0 km. | 74 km.W from Iquique |
| 05:02:49 | 5.0 | 10.0 km. | 115 km. WNW from Iquique |
| 06:04:10 | 5.1 | 10.0 km. | 70 km. WNW from Iquique |
| 11:07:30 | 5.4 | 10.0 km. | 92 km. WNW from Iquique |
| 19:45:50 | 5.4 | 21.2 km. | 43 km.WNW from Iquique |

Fig.3 shows the localization of the earthquake source and the location of the aftershocks in the next 2 days after the
main shock and data from the tide gauge in Patache.


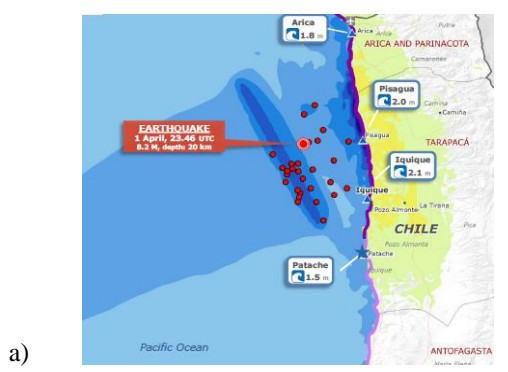

a)

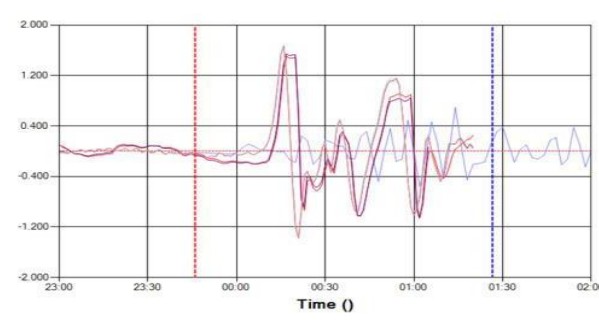

b)

**Fig. 3. a) Localization of the earthquake source and the location of aftershocks in the next 2 days after the main shock; b) data from a tide gauge in Patache (Internet resource ECDM_20140402_ Chile_Earthquake 2014).**

Fig.4 shows the shape of the seismic source in the simulation of this earthquake in two scenarios, selected according to the direction of seismic activity. For the numerical simulation of a tsunami from a seismic source localized along the Chilean coast, was used a part of basin Pacific Ocean in a square of $60^0$-$90^0$ W and $10^0$-$60^0$ S.

To estimate the size of the tsunami source and the height of the wave offset in the source, the formulas of Iida and Wells were used. (Wells and Coppersmith 1994), (Voltsinger et al. 1989).

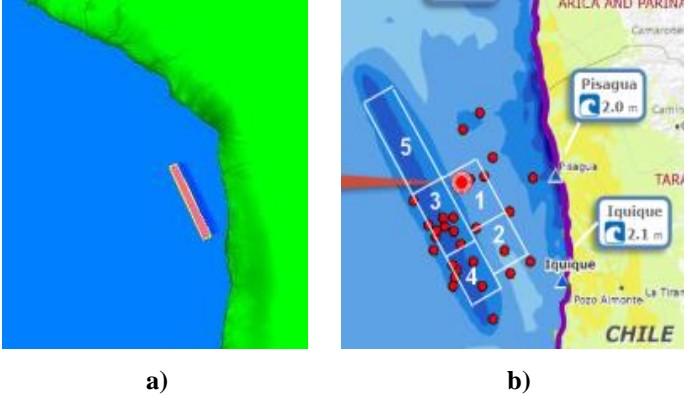

a)                                        b)

**Fig. 4. The shape of the earthquake source of April 1, 2014 for two computation scenarios: a) two-block seismic source; b) five-block seismic source. (Internet resource ECDM_20140402_ Chile_Earthquake 2014).**

To simulate the generation and propagation of a tsunami, for the realization of an earthquake there were considered two scenarios with two- and five-block sources. The main indicators of the adequacy of the simulation were selected tide-gauge records of European Commission modeling of disasters, which is freely available.


**Table 3. Characteristics of simulated sources.**

|  | Double block source | Five-block source |
|---|---|---|
| Coordinates source | (288.42; -21.25) – (286.58; -19) – (287.11; -18.5) – (289.21; -20.75) | (289.166; -20.066) – (288.7; -19.166) – (288.866; -19.1) – (289.133; -19.633) – (289.333; -19.533) – (289.6; -20.1) – (289.3; -19.9) – (289.466; -20.3) |
| Full time uplift | 60 sec | 320 sec |
| Vertical offsets size | 2, 6 m. | -0.3 – 2 m. |

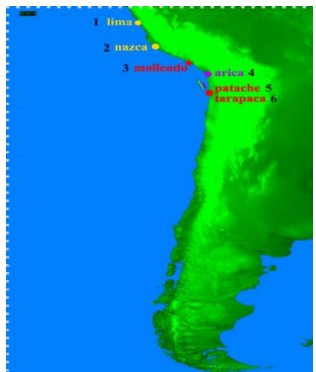

**Fig.5. Location of virtual tide gauges and seismic sources used in computations for Scenario 1 and Scenario 2. (Gebco Digital Atlas, electronic resource, 2018).**

_Scenario 1_. For scenario 1 we consider a bipolar elongated source of $125.3 \times 250.6$ km in size, oriented downward to the coast (Fig. 4a). The area of the source is 31000 km$^2$, with a height of maximum vertical lift of 9.1 m. (Table 3). In Fig.5 points are given on a 10-meter isobath, where virtual tide gauges were put up for calculation.

The results of computations in the Scenario 1 for the maximum wave height, for the height of the first wave, for the minimum level shift for the 10-meter isobath, are given in Table 4.



**Table 4. Computation results for the Scenario 1.**

| Points | max height | first wave height | min shift |
|---|---|---|---|
| **Patache (70.2°W 19.8°S)** | **7.69** | **7.69** | **3.1** |
| **Mollendo (72.1°W 17.05°S)** | **3.065** | **2,19** | **1.3** |
| **Nazca (75.5°W 15°S)** | **0.7** | **0.17** | **0.4** |
| **Lima  (77.13°W -12.08°S)** | **1.1** | **0.14** | **0.35** |
| **Tarapacá   (70.168°W 20.845°S)** | **6.61** | **-2.19** | **4** |
| **Arica  (70.34°W 18.5°S)** | **6.95** | **-3.27** | **3.5** |

In Fig.6 four positions of the wave fronts are given for the generation of an elongated bipolar source for Scenario 1 (Fig.4.a).

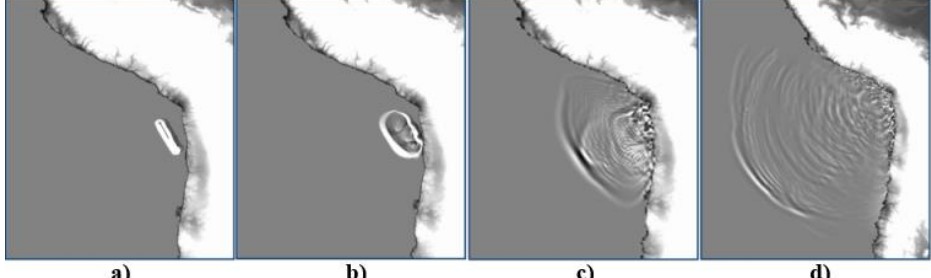

**Fig.6. The generation of a tsunami source by a two-block seismic source in the computation of Scenario 1 for the time moments: a) 30; b) 60; c)10000; d)300000 sec. (based on the Gebco bathymetry 2018)**

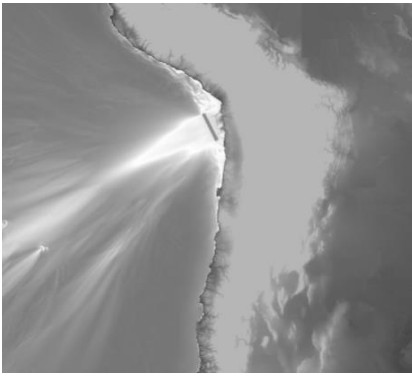

**Fig.7. Distribution of maximum wave heights for a part of this water area according to the results of numerical simulation under Scenario 1. (based on the Gebco bathymetry 2018)**



***Scenario 2***. For a more detailed description of the process, it is necessary to select the shape of the source corresponding to the direction of the seismic intensity of the process (Fig.3a). For this, a five-block source shape was chosen, which most adequately corresponds to the aftershock process in a given earthquake (Table 3, five-block source). When considering the second scenario, we proceeded from the condition of equality of the areas of the surfaces of both sources at the initial time moment. If we take into account the localization and time of aftershocks, then the shape of the source will have a more complex appearance (Fig.4b). According to the seismic process in the source, the movement of the keyboard blocks is defined in Table 5.

**Table 5. Characteristics of process for Scenario 2.**

| Number of blocks | 1 | 2 | 3 | 4 | 5 |
|---|---|---|---|---|---|
| **Bigining uplift (sec)** | 0 | 40 | 40 | 300 | 90 |
| **Finishing of uplift (sec)** | 10 | 100 | 100 | 320 | 120 |
| **Uplift height (m)** | 2.5 | -0.2 | -0.3 | 1 | 2 |

Fig.8 shows four time points for the generation and propagation of a tsunami wave in the implementation of Scenario 2.

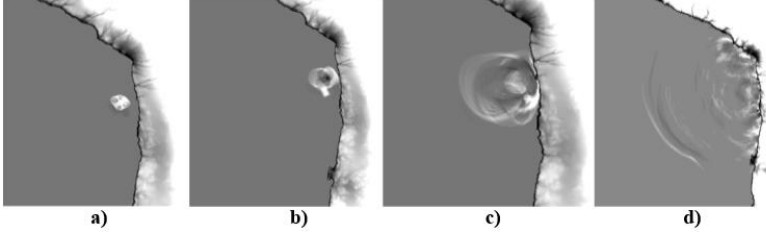

**Fig.8. Tsunami source generation and wave propagation in Scenario 2 for 4 time points: a) 30; b) 60; c) 10000; d) 300000 sec. (based on the Gebco bathymetry 2018)**

The simulation results are shown in Table 6.

**Table 6. Computation results for the Scenario 2**

| Points | max height | first wave height | min shift |
|---|---|---|---|
| Patache (70.2°W 19.8°S) | 2.24 | 2.24 | 1 |
| Mollendo (72.1°W 17.05°S) | 0.24 | 2,19 | 0.19 |
| Nazca (75.5°W 15°S) | 0.27 | 0.01 | 0.1 |
| Lima (77.13°W -12.08°S) | 0.18 | 1.43 | 0.03 |
| Tarapaca (70.168°W 20.845°S) | 2.02 | 0.2 | 1.8 |
| Arica (70.34°W 18.5°S) | 2.93 | | 0.9 |





**3. Analysis of the spectral characteristics of the wave field during a tsunami 01.04.2014 near the Chilean coast.**

Using the results of our computations of the generation and propagation of tsunami waves in a limited water area of the

255 Pacific Ocean near the northern part of the Chilean coast, a one-dimensional and wavelet spectral analysis was carried out and

the spectral characteristics of the wave process during a tsunami on April 01, 2014 in the near-field zone of the Chilean coast

were obtained. All points were calculated using the discrete Fourier transform, where the spectral density of the signal can be

presented as (M Kovacevic and J Goyal 2013), (Mallat 1999):

$$X(j\omega) = \sum_{n=0}^{N-1} x(n)\exp(-j\omega n),$$

(1)

where $x(n)$ is the discrete time signal, $N$ is the number of samples in the signal, $\omega = 2\pi f T$ is the normalized circular

frequency, $f$ is the frequency in Hertz *(Hz)*, $T$ is the sampling interval in seconds.

The graphs show the spectral density modules in decibels (dB):

$$L(\omega) = 20\lg(|X(j\omega)|)$$

(2)

The signal energy is written as follows:

$$E_x = \sum_{n=0}^{N-1} x^2(n).$$

(3)

Fig.9 shows the seismic source, and Fig.10, 11 show tide-gauge records and wavelet spectrograms of wave fields from near-

field sources corresponding to scenarios 1 and 2. The considered points are shown in the direction from the south of Tarapaca

to the north to Lima (Fig. 9).

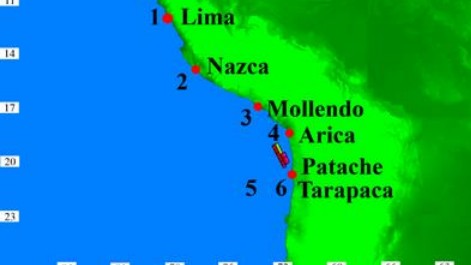

**Fig.9. The location of the points in which the virtual tide gauges were "exposed". (Gebco Digital Atlas, electronic resource, 2018).**

When considering both scenarios, we proceeded from the condition of equality of the areas of the surfaces of both sources at

280 the initial time moment. Since the first scenario is based on a two-block longitudinal model, when the block is oriented towards

the coast downwards, it is obvious that in most points of the nearest coast, a depression wave will be observed first. As it is

known from hydrophysics, (Pelinovsky & Mazova 1992) if a wave is preceded by a negative phase, then the next wave can





be significantly increased, as observed on tide-gauge records at those points where virtual tide gauges were exposed. If we take into account the localization and time of aftershocks, then the shape of the source will have a more complex form (Fig.4b).

According to the seismic process in the source, the movement of the keyboard blocks is defined in Table 5.

Fig.10, 11 show the comparison of wavelet spectrograms for both scenarios. It is well seen that the character of the distribution of low-frequency components for Tarapaca is essentially different. While for a two-block elongated source, the areas of greatest intensity (up to 35 dB) are in the low-frequency range and extended in time, but for a 5-block source, these areas are of lesser intensity (up to 20 cycle per hour (cph)), are more localized in time and shift in the frequency range up to 12 cph (which

corresponds to 8-min wave period). In p. Patache, as compared with the first scenario (Fig. 4.8b), the areas of greatest intensity (up to 20 dB) are distributed almost evenly throughout the entire time interval.

**First scenario**             **Second scenario**

**Tarapaca 1**             **Tarapaca 2**

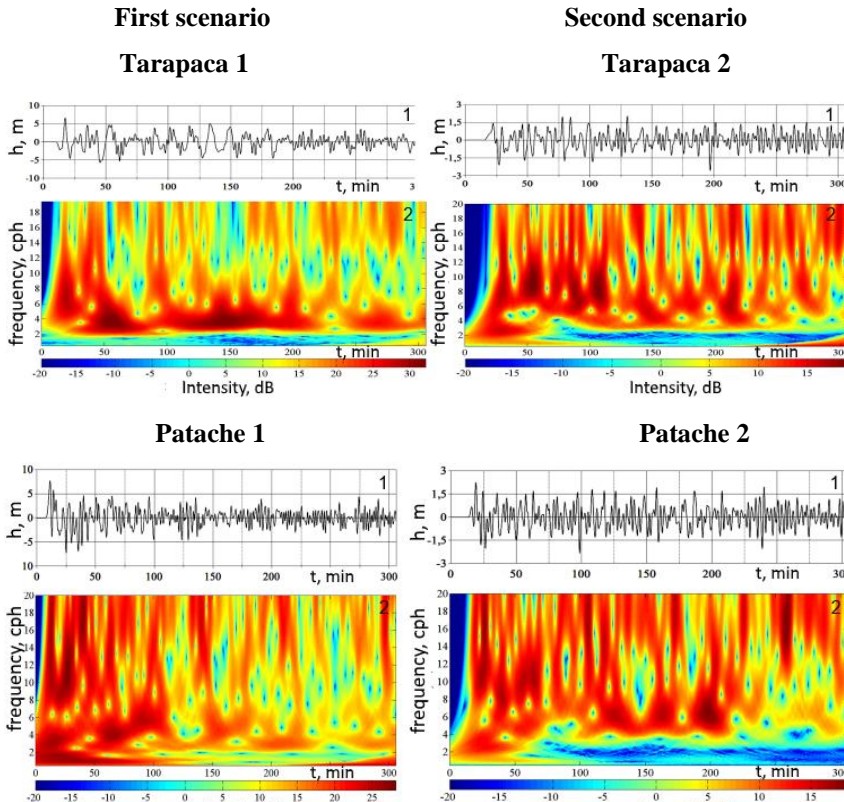

**Patache 1**             **Patache 2**




**Arica 1**            **Arica 2**

**Fig.10. Wave characteristics: (1) tide gauge record; (2) wavelet spectrogram for the points: Tarapaca, Patache, Arica.**

In p.Arica, the areas of low-frequency components are well pronounced at 4 time intervals: 25-50 min; 75-90 min; 95-110
310 min; 170-185 min with an intensity of up to 25 dB, which exceeds the intensity of the wave process in the implementation of
the first scenario. The regions are localized in the frequency range of 5-10 cph, which corresponds to 6-12 min waves. It is
clearly seen that the intensity of the wave process under implementation of the second scenario has increased essentially, which
seems to be associated with possible resonant effects with appropriate keyboard blocks movement in the source and a
characteristic curved coastline. Thus, it can be assumed that catastrophic events in the p. Arica region may be due to the
315 complex nature of the seismic source during a concrete earthquake.

**Mollendo 1**          **Mollendo 2**

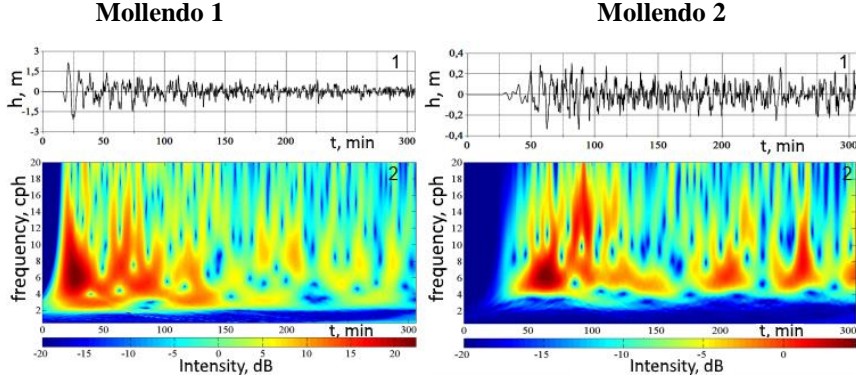



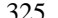

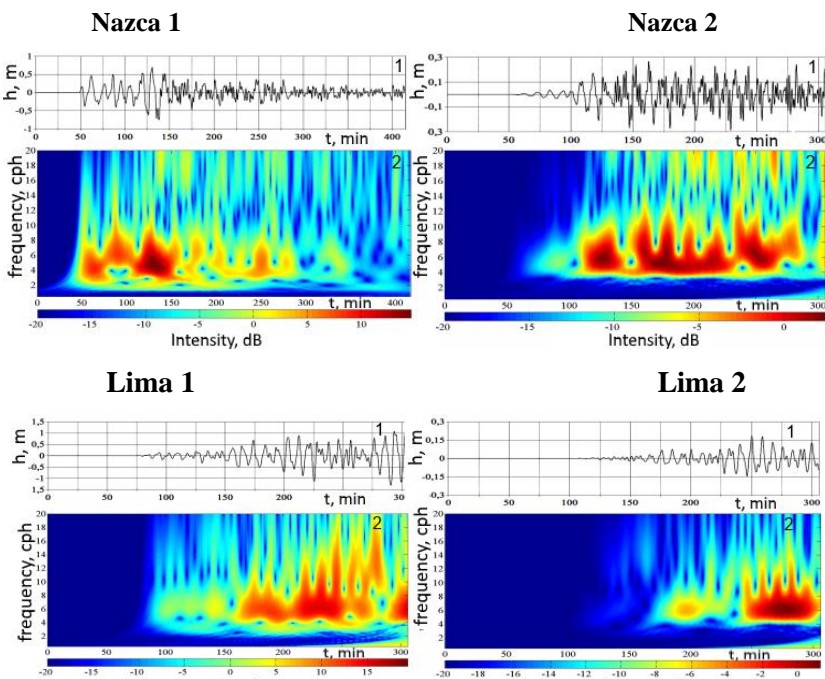

**Fig.11. Wave characteristics: (1) tide gauge record; (2) wavelet spectrogram for the points:   Mollendo, Nasca, Lima.**

At the points of Nazca and Lima, due to the interaction of a complex wave field with the peculiarities of the coastal relief, features of a shift over a time interval are observed, with a significant decrease in the intensity of the wave process during the propagation of a wave to northern Chile.

Thus, a numerical simulation of the last catastrophic tsunami with a seismic source, localized near the northern coast of Chile, performed, demonstrates that taking into account the complex structure of the seismic source allows us to describe a number

of effects in the near-field zone that are difficult, and sometimes impossible, to explain using a simplified model seismic source.

## Conclusion

Based on numerical simulation, a study of the generation and propagation of tsunami waves during the strongest earthquakes near the coast of Chile for the earthquake of April 1, 2014, allows us to estimate the maximum run-up heights throughout the

coast of the estimated water area. The tsunami caused real damage in many areas of the Pacific coast of northern Chile during the 2014 earthquake. Preliminary results obtained in this work allowed us to obtain additional data on the size, direction and intensity of seismic sources of the earthquake in Chile on April 1, 2014 on the parameters caused by it destructive tsunami in the Pacific. For numerical simulations, a keyboard model of earthquake sources was used, developed on the basis of additional data on real-time monitoring of seismic energy release peaks. The conducted numerical simulation of the source with the

division into keyboard blocks made it possible to study the features of the wave field formation both in the near-field and in



the far-field areas. Of particular note is the fact that the preliminary forecast given in (Mazova and Ramirez 1999) was in fact justified up to the absence of significant material damage on the coast and the preservation of human lives, in contrast to the earthquake of 1868-1877, the comparison with which was expected in the first and second decades this era.

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
