# Peer review of "Evidence of preliminary prognosis of appearance of catastrophic earthquake and strong tsunami in the region of Tarapacá, Chile"

_Natural Hazards and Earth System Sciences, 2019_

## Referee Comment (RC1) · Anonymous Referee #1 · 18 Oct 2019

Evaluation report:

General comments

The quality of discussed paper is good and meets the standards of the NHEES. My general comment is that there are several positive issues in the paper, especially regarding the investigation (numerical simulations of the earthquake and tsunami of 1st April, 2014, Chile off coast). Another strong positive approach is related to the detail spectral analysis, which is not very frequently used in tsunami papers. The investigated scenarios (1 and 2) add proves to the main hypothesis of the authors.

There are some individual scientific questions/issues as specific comments to the pa-

per: A) in Line 24 two important papers about the papers considering the stochastic prediction must be added (as well as in the References list): I. Papratilov, M. Velikova, B. Ranguelov  E. Spassov, 2011. Earthquake Prediction Stochastic Models – a Sofware using Matlab Algorithms. Application to the Chile Subduction Zone., Proc. 6th Congress of Balkan Geophysical Society - Budapest, Hungary, 3-6th October 2011. pp1-5.

Ranguelov B., I. Papratilov, M. Velikova, E. Spassov., 2011. A STOCHASTIC MODEL FOR PREDICTION THE OCCURRENCE OF STRONG EARTHQUAKES (M>7.0) IN THE CHILE SEISMOGENIC AREA., Ann. of MG University, Vol. 54, Part I, Geology and Geophysics., p. 173-176. ISSN 1312-1820

B) Line 40 - According to modern concepts of geotectonics (Lobkovsky L1988), (Lobkovsky L et al.  2004 ) , there are two types of subduction zones: the Chilean type and the Mariana type.  The Chilean type is characterized by a deep-sea trench and a strong coupling between the continental and oceanic lithospheric plates during the subduction process, and seismic activity in the Chilean-type subduction zones leads to strong tsunamis. Examples of Chilean-type subduction zones are the areas of the Kuril-Kamchatka and Japanese deep-sea trenches. . ...

Then the authors mentioned the structure and specifics of the Chilean type.  It is essential to do the same about Mariana type with 1-2 sentences.

C) It is really important somewhere after line 250 to put a short explanation regarding the importance of the refraction to the tsunami energy distribution; moreover such effects are clearly visible on fig. 6, 7 and 8. A sentence like: "It is important to note that the refraction play the significant part of the tsunami energy distribution and focused it to the nearest coast (i.e. Chielan)". This will improve the quality of the paper.

D) CONCLUSIONS - It is important to write 1-2 sentences in CONCLUSIONS, regarding the spectral analysis preformed.

- It is also preferable to put the last sentence from paragraph 3. Analysis . . .to CON-CLUSIONS (i.e. line 333 and below – "Thus, a numerical simulation of the last catastrophic tsunami with a seismic source, localized near the northern coast of Chile, performed, demonstrates that taking into account the complex structure of the seismic source allows us to describe a number of effects in the near-field zone that are difficult, and sometimes impossible, to explain using a simplified model seismic source.") to go to the CONCLUSIONS

For the "technical corrections" it is better to correct:

Line 70 – "Entire blocks of cities were washed away and destroyed. . .

It is more accurate to write instead of "cities" – "villages" (or inhabitant areas) as authors prefer.

Line 257 . "All points. . ." is incorrect. . . It is incorrect. "All points. . ." , better to write "The values of all points. . ."

Table 5 (second row) Bigining uplift. . .better to write "initial" (or beginning, if authors prefer)

Line 330 At the points of Nazca and Lima, due to the interaction of a complex wave field with the peculiarities of the coastal AND BOTTOM relief. . .,

It is important to add "and bottom"

Please also note the supplement to this comment:
https://www.nat-hazards-earth-syst-sci-discuss.net/nhess-2019-278/nhess-2019-278-RC1-supplement.pdf

**Supplement:**

Questionnaire of NHEES filled by referee:

1. Does the paper address relevant scientific and/or technical questions within the scope of NHESS?

Yes, the paper covered the scientific topics of NHEES.

2. Does the paper present new data and/or novel concepts, ideas, tools, methods or results?

The paper presents new data and results related to the earthquake and tsunami of 1st April, 2014 and investigate in details the selected scenarios about these disastrous events. .

3. Are these up to international standards?

The international standards of numerical simulations and spectral analysis are rather wide to these topics, so the paper is clearly attached to them.

4. Are the scientific methods and assumptions valid and outlined clearly?

Yes, all scientific methods and assumptions are valid and presented clearly, by text, figures, tables and results.

5. Are the results sufficient to support the interpretations and the conclusions?

The results and conclusions support the interpretations and outcomes and are verified by the presented scenarios of the investigated tsunamis.

6. Does the author reach substantial conclusions?

The authors reach substantial conclusions in the scope of the paper related to the results and interpretations.

7. Is the description of the data used, the methods used, the experiments and calculations made, and the results obtained sufficiently complete and accurate to allow their reproduction by fellow scientists (traceability of results)?

Yes, the traceability of results is sufficiently complete and accurate and allows the reproduction by the specialists. This is supported by the description of the algorithm, data and methods of calculation.

8. Does the title clearly and unambiguously reflect the contents of the paper?

I prefer a title as: **Preliminary prognosis of catastrophic earthquake and detailed study of the generated strong tsunami in the region of Tarapacá, Chile**

But the final decision is up to the authors.

9. Does the abstract provide a concise, complete and unambiguous summary of the work done and the results obtained?

Instead Line 21 "Thus, the evidences, presented in this work, support preliminary prognosis made by authors in 1999."

Will be more correct and informative according to the texts of the paper to write:

Thus, the investigations done, the interpretations and the results obtained presented in this work, support earlier prognosis made by authors in 1999.

10. Are the title and the abstract pertinent, and easy to understand to a wide and diversified audience?

The title and abstract are targeted to the professionals, but are understandable as well as to the wider public

11. Are mathematical formulae, symbols, abbreviations and units correctly defined and used? If the formulae, symbols or abbreviations are numerous, are there tables or appendixes listing them?

The mathematics and symbols are clear and the tables are appropriate listed.

12. Is the size, quality and readability of each figure adequate to the type and quantity of data presented?

The size and quality of figures is representative to all data presented.

13. Does the author give proper credit to previous and/or related work, and does he/she indicate clearly his/her own contribution?

Yes, it is. Some additional references are necessary to include especially to the stochastic earthquake prediction techniques (for example see p. 18)

14. Are the number and quality of the references appropriate?

Mostly the references are cited appropriately. Some adds are recommended.

15. Are the references accessible by fellow scientists?

Most of the references are accessible to the scientific community.

16. Is the overall presentation well structured, clear and easy to understand by a wide and general audience?

Yes, especially by the people working in the field of the tsunami numerical modeling.

17. Is the length of the paper adequate, too long or too short?

The length is normal.

18. Is there any part of the paper (title, abstract, main text, formulae, symbols, figures and their captions, tables, list of references, appendixes) that needs to be clarified, reduced, added, combined, or eliminated?

In the list of References is recommended to add some publications which are important to the completeness of the list:

I. Papratilov, M. Velikova, B. Ranguelov & E. Spassov, 2011. Earthquake Prediction Stochastic Models – a Sofware using Matlab Algorithms. Application to the Chile Subduction Zone., Proc. 6th Congress of Balkan Geophysical Society - Budapest, Hungary, 3-6$^{th}$ October 2011. 5pp. (on CD)

Ranguelov B., I. Papratilov, M. Velikova, E. Spassov., 2011. A STOCHASTIC MODEL FOR PREDICTION THE OCCURRENCE OF STRONG EARTHQUAKES (M>7.0) IN THE CHILE SEISMOGENIC AREA., Ann. of M&G University, Vol. 54, Part I, Geology and Geophysics., p. 173-176. ISSN 1312-1820

19. Is the technical language precise and understandable by fellow scientists?

The technical and scientific language is understandable by the specialized seismological and tsunami professionals auditory.

20. Is the English language of good quality, fluent, simple and easy to read and understand by a wide and diversified audience?

I'm not a professional linguist, so it is difficult to me to asses the quality of English language used. There are some minor corrections to be done.

---

## Referee Comment (RC2) · Anonymous Referee #2 · 20 Dec 2019

The present article apparently attempts to show that the earthquake and tsunami that took place on April 1st, 2014 near Pisagua, Chile had been predicted by an earlier article of some of the authors.

I consciously stress that they \*apparently\* intend to do so, because the article itself lacks clarity on this regard and this could be only inferred from the title and a passing comment on the later stages of the article. It is hard to find other evidence for such claim other than a brief reference to earlier work. Whether these correlate is not established at all bar the fact that an earthquake occurred in a seismic gap.

Upon reading the article, moreover, it is difficult to understand its actual aim and objectives. The article not only lacks a clear structure and line of reasoning (and leaving aside the very poor english grammar), but constantly offers a wide range of information that does not relate to the main topic, which obfuscates the reading. While doing this, sometimes elements are omitted without explanation, which is even more confusing. For example, when reviewing the history of Chilean seismicity, events such as those of 1922 and 1943 are omitted despite its importance. Tables and figures appear rather haphazardly and very often they have no explicit relation to the text.

In addition to these formal aspects, there are a few fundamental issues. The referencing is mostly dated and self-referencing. This would not be problem per se, but in doing so the authors ignore a large body of research published since the occurrence of the earthquake and tsunami of interest. That research have done meaningful progress in characterizing these events. Several of these works touch upon topics that are closely related to the ones of the present article, and they do so with greater detail. To mention just a few:

Seismic source inversion Chen, K.; Babeyko, A.; Hoechner, A. & Ge, M. (2016), 'Comparing source inversion techniques for GPS-based local tsunami forecasting: A case study for the April 2014 M8.1 Iquique, Chile, earthquake', Geophysical Research Letters 43(7), 3186–3192.

Gusman, A. R.; Murotani, S.; Satake, K.; Heidarzadeh, M.; Gunawan, E.; Watada, S. & Schurr, B. (2015), 'Fault slip distribution of the 2014 Iquique, Chile, earthquake estimated from ocean-wide tsunami waveforms and GPS data', Geophysical Research Letters 42.

Schurr, B.; Asch, G.; Hainzl, S.; Bedford, J.; Hoechner, A.; Palo, M.; Wang, R.; Moreno, M.; Bartsch, M.; Zhang, Y.; Oncken, O.; Tilmann, F.; Dahm, T.; Victor, P.; Barrientos, S. & Vilotte, J.-P. (2014), 'Gradual unlocking of plate boundary controlled initiation of the 2014 Iquique earthquake', Nature 512(7514), 299–302.

Yagi, Y.; Okuwaki, R.; Enescu, B.; Hirano, S.; Yamagami, Y.; Endo, S. & Komoro, T.

(2014), 'Rupture process of the 2014 Iquique Chile Earthquake in relation with the foreshock activity', Geophysical Research Letters 41, n/a–n/a.

Tsunami Modeling: Calisto, I.; Ortega, M. & Miller, M. (2015), 'Observed and modeled tsunami signals compared by using different rupture models of the April 1, 2014, Iquique earthquake', Natural Hazards, 1-12.

An, C.; Sepúlveda, I. & Liu, P. L.-F. (2014), 'Tsunami Source and Its Validation of the 2014 Iquique, Chile Earthquake', Geophysical Research Letters 41(11), 3988–3994.

Omira, R.; Baptista, M. A. & Lisboa, F. (2016), 'Tsunami Characteristics Along the Peru–Chile Trench: Analysis of the 2015 Mw8.3 Illapel, the 2014 Mw8.2 Iquique and the 2010 Mw8.8 Maule Tsunamis in the Near-field', Pure and Applied Geophysics, 1–15.

Tsunami modeling and resonant activity in the area

Catalán, P. A.; Aránguiz, R.; González, G.; Tomita, T.; Cienfuegos, R.; González, J.; Shrivastava, M. N.; Kumagai, K.; Mokrani, C.; Cortés, P. & Gubler, A. (2015), 'The 1 April 2014 Pisagua tsunami: Observations and modeling', Geophysical Research Letters 42(8), 2918–2925.

Cortés, P.; Catalán, P. A.; Aránguiz, R. & Belloti, G. (2017), 'Tsunami and shelf resonance on the northern Chile coast', Journal of Geophysical Research 122(9), 7364-7379.

It is therefore unclear why they are not considered at all. This poses a serious problem for the article as it adds no value to the current state of knowledge.

There are fundamental methodological errors as well. They use appear strike angle that bears no correlation with the physical configuration of the area under study, although it is not possible to know for sure from the data as presented, since for each block only the location and displacement are given. Moreover, they use a sparse set of sea surface elevation data to validate the modeling, omitting time series very close

to the source such as that of Iquique and Pisagua. How it is possible to expect a valid source model omitting such relevant data?

With all these considerations, I consider that there is no alternative other than rejection of the manuscript.

---

## Author Comment (AC1) · 4 Feb 2020

I sent you our author comments to comments from both Referees.

Referee 1 examined our work in detail and made comments, with which we largely agree. The Referee 1, obviously, well represents the computed method described in our paper. Therefore, his comments were "line by line", i.e. detailed, which can also be answered essentially. But, with a number of comments by Referee 1, we do not agree, as I write in detail in my answers in Attachment 2.

Referee 2 did a great work comparing our paper with existing publications on this event.

[Figure]

However, unfortunately, he is obviously had little to do with the work on the application of the keyboard model of the earthquake to the description of catastrophic earthquakes, its justification, and the further widespread use of this technique to model complex catastrophic events accompanied by tsunamis. This model was successfully applied to the Kuril-Kamchatka subduction zone, which almost accurately indicated (first in the world!) the location of the epicenters of the three strongest tsunamigenic earthquakes, September 30, 2006, November 15, 2006, and January 13, 2007 that followed in the same region six months after the work was published (Lobkovsky, L.I., Mazova, R. Kh. et al., (Doklady RAS, 2006)). Since the paper (Mazova R.Kh. and Ramirez J.F., 1999) provides conclusions of the authors on the similarity of the continental slope of the deep sea trench near of northwestern part of the Chilean coast and Kuril-Kamchatka area, in this paper the keyboard model of the earthquake was also used. Since this model is widely published (see, e.g. [1,2] below), we did not see the need to go into its details, but considered only its use for modeling this earthquake. Unfortunately, the Referee 2 also "did not notice" the study of this process by the method of spectral analysis, which is rarely enough, in such a volume, used to analyze the wave characteristics of the tsunami process. In our answers, we tried to briefly explain some of the errancies of our colleague (see Attachment 3).

I do hope that our answers to comments from Referees will be convincing for you to consider the revised version of our paper for possible publication.

1. L.Lobkovsky, I.Garagash, B.Baranov, R.Mazova, N.Baranova. Modeling Features of Both the Rupture Process and the Local Tsunami Wave Field from the 2011 Tohoku Earthquake // Pure Appl. Geophys. (2017). V.174, p. 3919-3938, doi:10.1007/s00024-017-1539-5 (March 2017) pp.1-20. 2. L.I.Lobkovsky, I.A.Garagash, R.Kh.Mazova, Numerical simulation of tsunami waves generated by the underwater landslide for the Northern Coast of the Black Sea (Dzhubga Area)// Geophysical J. Int., V.218, Iss.2, Aug 2019, p. 1298–1306, http//doi.org/10.1093/gji/ggz221.

Sincerely yours, on behalf of all co-authors, Raissa Mazova

Please also note the supplement to this comment:
https://www.nat-hazards-earth-syst-sci-discuss.net/nhess-2019-278/nhess-2019-278-AC1-supplement.pdf

———————————————————

[Figure]

**Supplement:**

**Authors Comments to Comments of Referee 1** (Mazova et al. "Evidence of …")

1. *A) in Line 24 two important papers about the papers considering the stochastic prediction must be added (as well as in the References list).*

**Answer:** 2 papers are added into the revised text of the manuscript.

a) Papratilov, M. Velikova, B. Ranguelov E. Spassov, 2011. Earthquake Prediction Stochastic Models – a Sofware using Matlab Algorithms. Application to the Chile Subduction Zone., Proc. 6th Congress of Balkan Geophysical Society - Budapest, Hungary, 3-6th October 2011. pp1-5.
b) Ranguelov B., I. Papratilov, M. Velikova, E. Spassov., 2011. A STOCHASTIC MODELFOR PREDICTION THE OCCURRENCE OF STRONG EARTHQUAKES (M>7.0) IN THE CHILE SEISMOGENIC AREA., Ann. of MG University, Vol. 54, Part I, Geology and Geophysics., p. 173-176. ISSN 1312-1820 B)

2. *B) Line 40 - …According to modern concepts of geotectonics (Lobkovsky L1988), (Lobkovsky L et al. 2004 ) , there are two types of subduction zones: the Chilean type and the Mariana type…*

**Answer:** Our work (Mazova R.Kh. and Ramirez J.F., 1999) provides conclusions of the authors on the similarity of the continental slope of the deep sea trench near of northwestern part of the Chilean coast and Kuril-Kamchatka area, therefore, in this work also used a keyblock model of the underwater earthquake. However, the Mariana Trench has a quite different structure and, in our opinion, it is not relevant to mention it in this work.

3. *C) It is really important somewhere after line 250 to put a short explanation regarding the importance of the refraction to the tsunami energy distribution; moreover such effects are clearly visible on fig. 6, 7 and 8. A sentence like: "It is important to note that the refraction play the significant part of the tsunami energy distribution and focused it to the nearest coast (i.e. Chielan)". This will improve the quality of the paper.*

**Answer:** Of course, one could write a lot about the influence of reflection effects on wave heights on the coast and consider each point specifically. However, it seems to us that we have described in sufficient detail all of the wave processes, including using spectral analysis of wave fields. Enlarging an already large article, apparently, does not make sense.

4. *D) CONCLUSIONS - It is important to write 1-2 sentences in CONCLUSIONS, regarding the spectral analysis preformed. - It is also preferable to put the last sentence from paragraph 3. Analysis . to CONCLUSIONS (i.e. line 333 and below – "Thus, a numerical simulation of the last catastrophic tsunami with a seismic source, localized near the northern coast of Chile, performed, demonstrates that taking into account the complex structure of the seismic source allows us to describe a number of effects in the near-field zone that are difficult, and sometimes impossible, to explain using a simplified model seismic source.") to go to the CONCLUSIONS*

**Answer:** Such corrections will be of course made in the revised version of the text.

5. *For the "technical corrections" it is better to correct: Line 70 – "Entire blocks of cities were washed away and destroyed. . . It is more accurate to write instead of "cities" – "villages" (or inhabitant areas) as authors prefer. Line 257 . "All points. . ." is incorrect. . . It is incorrect. "All points. . ." , better to write "The values of all points. . ." Table 5 (second row) Bigining uplift. . .better to write "initial" (or beginning, if authors prefer) Line 330 At the points of Nazca and Lima, due to the interaction of a complex wave field with the peculiarities of the coastal AND BOTTOM relief. . .,*

**Answer:** These corrections will be of course made in the revised version of manuscript.

---

## Author Comment (AC2) · 4 Feb 2020

Dear Editor, I sent you our author comments to comments from both Referees. Referee 1 examined our work in detail and made comments, with which we largely agree. The Referee 1, obviously, well represents the computed method described in our paper. Therefore, his comments were "line by line", i.e. detailed, which can also be answered essentially. But, with a number of comments by Referee 1, we do not agree, as I write in detail in my answers in Attachment 2. Referee 2 did a great work comparing our paper with existing publications on this event. However, unfortunately, he is obviously had little to do with the work on the application of the

keyboard model of the earthquake to the description of catastrophic earthquakes, its justification, and the further widespread use of this technique to model complex catastrophic events accompanied by tsunamis. This model was successfully applied to the Kuril-Kamchatka subduction zone, which almost accurately indicated (first in the world!) the location of the epicenters of the three strongest tsunamigenic earthquakes, September 30, 2006, November 15, 2006, and January 13, 2007 that followed in the same region six months after the work was published (Lobkovsky, L.I., Mazova, R. Kh. et al., (Doklady RAS, 2006)). Since the paper (Mazova R.Kh. and Ramirez J.F., 1999) provides conclusions of the authors on the similarity of the continental slope of the deep sea trench near of northwestern part of the Chilean coast and Kuril-Kamchatka area, in this paper the keyboard model of the earthquake was also used. Since this model is widely published (see, e.g. [1,2] below), we did not see the need to go into its details, but considered only its use for modeling this earthquake. Unfortunately, the Referee 2 also "did not notice" the study of this process by the method of spectral analysis, which is rarely enough, in such a volume, used to analyze the wave characteristics of the tsunami process. In our answers, we tried to briefly explain some of the errancies of our colleague (see Attachment 3). I do hope that our answers to comments from Referees will be convincing for you to consider the revised version of our paper for possible publication. 1. L.Lobkovsky, I.Garagash, B.Baranov, R.Mazova, N.Baranova. Modeling Features of Both the Rupture Process and the Local Tsunami Wave Field from the 2011 Tohoku Earthquake // Pure Appl. Geophys. (2017). V.174, p. 3919-3938, doi:10.1007/s00024-017-1539-5 (March 2017) pp.1-20. 2. L.I.Lobkovsky, I.A.Garagash, R.Kh.Mazova, Numerical simulation of tsunami waves generated by the underwater landslide for the Northern Coast of the Black Sea (Dzhubga Area)// Geophysical J. Int., V.218, Iss.2, Aug 2019, p. 1298–1306, http//doi.org/10.1093/gji/ggz221. Sincerely yours, on behalf of all co-authors, Raissa Mazova

Please also note the supplement to this comment:

https://www.nat-hazards-earth-syst-sci-discuss.net/nhess-2019-278/nhess-2019-278-AC2-supplement.pdf

**Supplement:**

**Authors Comments to Comments of Referee 2 (Mazova et al. "Evidence of …" )**

1. ***The present article apparently attempts to show that the earthquake and tsunami that took place on April 1st, 2014 near Pisagua, Chile had been predicted by an earlier article of some of the authors***.

**Answer:** In 1999, in a joint Chilean-Russian study of catastrophic earthquakes and tsunamis near the Chilean coast (see, e.g., Mazova R.Kh. and Ramirez J.F. "*Tsunami waves with an initial negative wave on the Chilean coast*" // Natural Hazards (1999)), it was proposed that a **catastrophic** earthquake similar to 1877 should occur over the next 10–20 years (i.e. in 2010-2020) of the **northwestern part** of the Chilean coast (see recurrence law in (Mazova R.Kh. and Ramirez J.F., 1999)). The present paper indicates that the **catastrophic** event of earthquake and tsunami that took place on April 1st, 2014 near Pisagua, Chile is supporting to such proposal of seismic gap for such event (100-150 years) presented in our earlier works (Mazova and Ramires, 1999; Mazova and Soloviev, 1995).

2. ***I consciously stress that they \*apparently\* intend to do so, because the article itself clarity on this regard and this could be only inferred from the title and a passing comment on lacks the later stages of the article***.

**Answer:** In this paper, unlike all cited by the Referee, the authors consider a dynamical keyboard model of an earthquake. This model was successfully applied to the Kuril-Kamchatka subduction zone, which almost accurately indicated (first in the world!) the location of the epicenters of the three strongest tsunamigenic earthquakes, September 30, 2006, November 15, 2006, and January 13, 2007 that followed in the same region six months after the work was published (Lobkovsky, L.I., Mazova, R. Kh. et al., Doklady RAS, 2006). Since the paper (Mazova R.Kh. and Ramirez J.F., 1999) provides conclusions of the authors on the similarity of the continental slope of the deep sea trench near of northwestern part of the Chilean coast and Kuril-Kamchatka area, in this paper the keyboard model of the earthquake was also used. It should be noted that this model was additionally confirmed recently with satellite geodesy data (Lobkovsky et al., 2019).

3. ***It is hard to find other evidence for such claim other than a brief reference to earlier work.***

**Answer:** In the paper, in addition to "a brief reference to earlier work" there are presented the number of other evidences supporting our proposal. We believe that our justification for the possibility of a **catastrophic** earthquake in the seismic gap of the middle Kuril Islands and its confirmation is sufficient to justify the possibility of a **catastrophic** earthquake and tsunami in the zone under consideration, based on the recurrence law for the Chilean coast, constructed in (Mazova R.Kh. and Ramirez J.F., 1999).

4. ***Whether these correlate is not established at all bar the fact that an earthquake occurred in a seismic gap***.

**Answer:** The work deals with the repetition of a **catastrophic** tsunamigenic earthquake of a similar magnitude in the same section of the water area after a certain period of time.

5. ***Upon reading the article, moreover, it is difficult to understand its actual aim and objectives.***

**Answer:** The "actual aim" of the paper is to indicate that a data and location of the event on April 1st, 2014 near Pisagua, Chile corresponds to predictions following from recurrence law constructed in our previous work (Mazova and Ramirez, 1999) (also, see above (points 1,2)). Also, a numerical simulation of this event was carried out within the framework of the keyboard model, which was compiled by the staff of the seismology laboratory of the Shirshov Institute of Oceanology of the Russian Academy of Sciences.

6. *The article not only lacks a clear structure and line of reasoning (andd leaving aside the very poor english grammar), but constantly offers a wide range of information that does not relate to the main topic, which obfuscates the reading*.

**Answer:** We agree that the structure of the paper needs to be made more understandable. The grammar of the English language will be adjusted, and unnecessary information, where possible, will be reduced.

7. *While doing this, sometimes elements are omitted without explanation, which is even more confusing. For example, when reviewing the history of Chilean seismicity, events such as those of 1922 and 1943 are omitted despite its importance.*

**Answer:** The work mainly deals with the *catastrophic* events of this century, and the Table shows the *catastrophic* events of only this century, the rationale for which is given in reference to the work (Mazova and Soloviev, 1994), which refers to seismic activity around the perimeter of the Pacific Ocean to be increased significantly by the end of the 20th and the beginning of the 21st centuries.

8. *Tables and figures appear rather haphazardly and very often they have no explicit relation to the text.*

**Answer:** It is not clear to me what Tables and figures the reviewer speaks of, I need to write in detail.

9. *The referencing is mostly dated and self-referencing*.

**Answer:** The authors of this paper refer not to dated references but to classical works that determine the essence of the problem (such works are relevant in any paper). As for a large number of different works devoted to this earthquake and tsunami, as a rule, any similar event causes a series of the same type of work, slightly differing in ideological orientation. The references to our papers are only those including our original method not yet available in other papers, and are given only when it is necessary.

10. *This would not be problem per se, but in doing so the authors ignore a large body of research published since the occurrence of the earthquake and tsunami of interest*.

**Answer:** Yes, we'll add a couple of papers to the list of references.

11. *Several of these works touch upon topics that are closely related to the ones of the present article, and they do so with greater detail.*

**Answer:** Yes, we'll add a couple of papers to the list of references.
1. Lay, T., H. Yue, E. E. Brodsky, and C. An (2014), The 1 April 2014 Iquique, Chile, Mw 8.1 earthquake rupture sequence, Geophys. Res. Lett., 41, doi:10.1002/2014GL060238.
2. Gusman, A. R., S. Murotani, K. Satake, M. Heidarzadeh, E. Gunawan, S. Watada, and B. Schurr (2015), Fault slip distribution of the 2014 Iquique, Chile, earthquake estimated from ocean-wide tsunami waveforms and GPS data, Geophys. Res. Lett., 42, 1053–1060, doi:10.1002/2014GL062604.

12. *It is therefore unclear why they are not considered at all*.

**Answer:** We know these works, however, for present work there is no special need to include them in the consideration, since each of these works has its own solution features and its own deficiencies in solving the problem (see, e.g. Chen et al., 2016).

13. *This poses a serious problem for the article as it adds no value to the current state of knowledge*.

**Answer:** The article gives a quite novel understanding of the problem of tsunami generation and propagation, since the dynamical structure of the keyboard seismic source in the subduction zone, which is characteristic for Chilean subduction zone, is important for this problem.

14. ***There are fundamental methodological errors as well***.
**Answer:** (see above).

15. ***They use appear strike angle that bears no correlation with the physical configuration of the area under study, although it is not possible to know for sure from the data as presented, since for each block only the location and displacement are given***.
**Answer:** The choice of the earthquake source, its shape and location, was based on accurate seismic data from the USGS National Earthquake Information Center, summarized by the staff of the seismology laboratory of the Shirshov Institute of Oceanology of the Russian Academy of Sciences. An analysis of the dynamical transition process of the formation of the distribution of displacements of the seabed shows that the processes occurring in the seismic source are ultimately converted to the dynamical component of the vertical displacement of the bottom. Therefore, taking into account all the characteristic parameters of an earthquake, we recalculate them into a vertical displacement (reverse or fault). In this case, due to the incompressibility of the water and hydrostatic pressure, a tsunami source is formed, and the wave height above the seismic source will be the same as the displacement in the source.

16. ***Moreover, they use a sparse set of sea surface elevation data to validate the modeling, omitting time series very close to the source such as that of Iquique and Pisagua. How it is possible to expect a valid source model omitting such relevant data?***
**Answer:** (see point 15).

17. ***How it is possible to expect a valid source model omitting such relevant data?***
**Answer:** (see point 15).